# Concurrent trajectories of part-time work and sickness absence: a longitudinal cohort study over 11 years among shift working hospital employees

Annina Ropponen [1,2] Jenni Ervasti [1] Mikko Härmä[1]

[1]Finnish Institute of Occupational Health, Helsinki, Finland
[2]Department of Clinical Neuroscience, Karolinska Institute, Stockholm, Sweden

**Correspondence to**
Dr Annina Ropponen;
annina.ropponen@ttl.fi

## ABSTRACT

**Objectives** To investigate the concurrent changes in part-time work and sickness absence (SA) in healthcare. Another aim was to investigate the role of age and sex on different concurrent trajectory groups.

**Design** Prospective cohort study.

**Setting** Public hospital districts (n=10) and cities (n=11) in Finland.

**Participants** Payroll-based objective working hour data of the healthcare sector in Finland for 28 969 employees in 2008–2019 were used. The final sample included those working shifts with 3 consecutive years of data and without baseline (≥14 days) SA.

**Primary outcomes** Part-time work (yes or no) and months of SA.

**Measures** Group-based trajectory modelling to identify concurrent changes in part-time work, and months of SA while controlling the time-variant amount of night work and multinomial regression models for relative risk (RR) with 95% CIs were used.

**Results** Four-group trajectory model was the best solution: group 1 (61.2%) with full-time work and no SA, group 2 (16.9%) with slowly increasing probability of part-time work and low but mildly increasing SA, group 3 (17.6%) with increasing part-time work and no SA, and group 4 (4.3%) with fluctuating, increasing part-time work and highest and increasing levels of SA. Men had a lower (RR 0.49–0.75) and older age groups had a higher likelihood (RRs 1.32–3.79) of belonging to trajectory groups 2–4.

**Conclusions** Most of the sample were in the trajectory group with full-time work and no SA. The probability of part-time work increased over time, linked with concurrent low increase or no SA. A minor group of employees had both an increased probability of part-time work and SA. Part-time work and other solutions might merit attention to promote sustainable working life among healthcare employees.

## BACKGROUND

The healthcare sector is facing major challenges due to population ageing, increased need for care and economic challenges combined with staff shortages. The ageing population affects both ensuring sufficient levels of nursing staff and the growing demands for the care of the ageing population.[1 2] The existing need for longer work careers[3] combined with increasing turnover rates emphasises the urgent need to understand working hours in association with work capacity and sustainable working life. This calls for an understanding of associations between working hours and work capacity and builds on sustainable work which is 'working and living conditions that are such that they support people in engaging and remaining in work throughout an extended working life'.[4] For example, in recent years, sickness absences (around 10% of all employed) have been increasing in Finland[5] and especially in healthcare.[6] In shift work, two working hour solutions may provide a possibility to decrease workload due to demanding working conditions or to arrange more time for care for small children or other dependents: an

---

**STRENGTHS AND LIMITATIONS OF THIS STUDY**

⇒ Payroll-based register data of working hours enabled objective assessment of part-time work and sickness absence.

⇒ A homogeneous sample of hospital employees working in specialised healthcare services or at primary healthcare, including for example, health centres and local hospitals for elderly and chronically ill people providing 24/7 care, or home care for the disabled.

⇒ Group-based trajectory model to identify concurrent trajectories of part-time work and mean number of sickness absence months/years for the follow-up from 2009 to 2019 to estimate a distinct pattern over time.

⇒ Lack of information on the reasons for part-time work and no information on diagnosis or part-time sickness absence.

employee may work part-time, or to change from night work to daytime work.[7 8] However, part-time work may also be health related, that is, being based on part-time sickness absence or disability pension.[9 10] Until now, studies on the role of part-time work concerning future long-time trajectories of sickness absence are lacking. As sickness absence increases costs both to the employers and the society, further understanding of the linkage with part-time work is needed.

Working part-time may, besides influencing income, have long-term consequences on career and pension.[11] In Finland, every fifth woman (13% of men) works part-time.[12] On the other hand, shorter working hours and increased time for recovery could lengthen the work careers.[13] Due to an emergent need to increase employment rates and promote sustainable working life among the working population,[14 15] longitudinal studies investigating the role of part-time work are important. Many studies have reported associations between individual sociodemographic, socioeconomic, health-related and work-related factors including part-time work for labour market exit due to sickness absence or disability pension.[16 17] However, until now, studies with longitudinal designs to investigate associations between part-time work and sustainable working life have mainly been based on survey data.[15 18–23]

This study hypothesised that due to ageing, older employees may choose part-time work due to decreased health (reflected in higher sickness absence) more often than younger employees. We also hypothesise that part-time work without increased sickness absence might be more frequent among younger employees than older ones, for example, due to the need to combine work and life while having small children or with studies. The possibility to combine part-time work with part-time sickness absence, that is, to shorten working hours when having health problems and while compensated by social benefits (ie, sickness absence benefits), may extend and/or maintain the working lives of the older employees.[24] However, part-time work and sickness absence may also both increase with age. Another aspect related to age is night work, that is, older employees may wish to work more during the day than at night.[25] Also, age-related trends in the associations of night work and health have been identified.[26 27] Therefore, based on the assumption that younger and older employees might react differently to night work,[28 29] one should control night work when investigating age-related differences in part-time work and sickness absence. This is relevant, especially in longitudinal settings in which working during the night may decrease due to the possibility of selecting more favourable working hours.[30 31]

This study aimed to investigate the concurrent changes in part-time work and sickness absence among employees in healthcare. Another aim was to investigate the role of age and sex on different concurrent trajectory groups.

## METHODS

This study was based on the ongoing Working Hours in the Finnish Public Sector Study[32 33] with payroll-based register data of working hours for 2008–2019. To ensure follow-up across years and to exclude those with exceptionally long absences (due to sickness, parental leave or other reasons) and very short working periods, we restricted the sample to those with ≥31 work shifts/year in 3 consecutive years during 2008–2019. Then, the sample was restricted to those without sickness absence (≥14 consequent days) at baseline in 2008. Last, only those with shift work contracts were included (figure 1). The final sample was 28 969 shift working employees in 10 hospital districts and 11 cities using Titania shift scheduling software. In Finland, the hospital districts (45% of the final sample) are responsible for specialised healthcare services providing both specialised outpatient and inpatient care. The healthcare employees in the cities (55% of the final sample) are those working at primary healthcare, including, for example, health centres and local hospitals for elderly and chronically ill people providing 24/7 care, or home care for the disabled.[33]

In this study, we used annual working hours and sickness absence data from 2008 to 2019 as has been described in detail before.[32 33] Part-time work was based on work contract information and coded yes or no for each year based on the annual median. Then, annual sickness absence of any length was measured in months/years for follow-up from 2009 to 2019. For the baseline 2008, the short sickness absences (≤3 days) were estimated. We also used the proportion of night work of all work shifts/year to estimate the time-varying effect of night work as a proxy of shift work in this study. Since ageing employees may change from night work to day work,[25 34 35] the proportion of night work was assumed to reflect this change, which in turn might affect the rate of part-time work. Baseline age (in 2008 and years categorised into <25 years of age, ≥25 and <40 years, ≥40 and <55 years, and >55 years based on distribution) and sex were used in the analyses.

### Patient and public involvement

Patients or the public were not involved in the design, or conduct, or reporting, or dissemination plans of our research.

### Statistical analyses

First, we calculated the descriptive characteristics of the study sample. Then, using a group-based multitrajectory model, we identified concurrent trajectories of part-time work and mean number of sickness absence months/years for the follow-up from 2009 to 2019 while accounting for night work as time-varying covariate.[36] The group-based multitrajectory method enables to identify groups of individuals (trajectory groups) that follow a distinct pattern over time. The linear polynomial model was applied. Bayesian information criterion (BIC), Akaike information criterion (AIC) and average posterior probability were used to determine the best-fitting model. Third, we

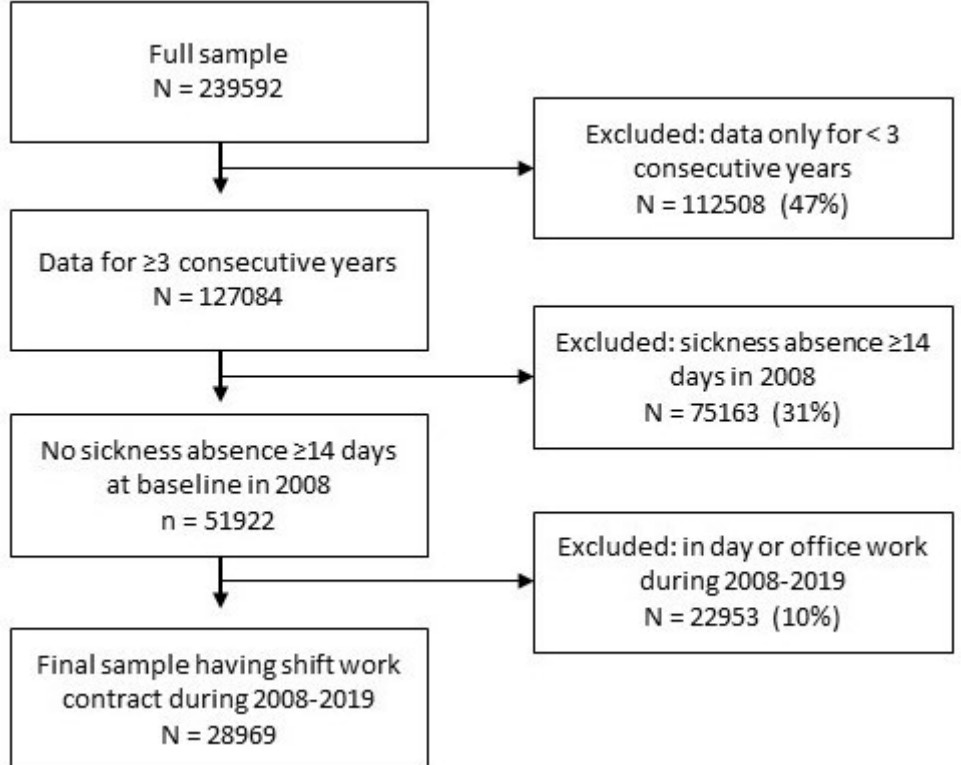

**Figure 1** Sample selection.

ran multinomial regression models for relative risk and 95% CIs using sex and age groups to predict the group membership for each trajectory group. All statistical analyses were conducted with Stata V.17.1 MP.

## RESULTS

At baseline in 2008, 7% of the final sample of 28 969 shift working employees worked part-time, the mean age was 39.8 years (SD 10.7), 92% were women and 34% had had at least one short sickness absence (≤3 days). The amount of night shifts was 8% of all shifts. During the follow-up (from 2009 to 2019), the mean number of sickness absence months varied from 0.3 months in 2009 to 0.6 months in 2018 (range 0–12 months).

A four-cluster model was chosen based on BIC and AIC values and parsimony for concurrent changes in part-time work and months of sickness absence (table 1).

The concurrent trajectories were (figure 2):
1. Group 1 (61.2% of the final sample) with full-time work and no sickness absence.
2. Group 2 (16.9) with slowly increasing probability of part-time work and low but mildly increasing sickness absence.
3. Group 3 (17.6%) with increasing part-time work and no sickness absence.
4. Group 4 (4.3%) with fluctuating but increasing part-time work and highest and increasing levels of sickness absence.

The regression models (table 2) showed that compared with the youngest age group (<25 years of age at the baseline), the older age groups had an increased likelihood of belonging to trajectory groups 2–4 (in comparison with trajectory group 1). Men had a lower likelihood of belonging to trajectory groups 2–4 than women (table 2).

In the online supplemental material, we present the concurrent trajectories without adjusting for time-variant

**Table 1** Goodness-of-fit statistics of group-based trajectory analysis models

| | Smallest group | | | | |
| | N | % | BIC | AIC | APP |
|---|---|---|---|---|---|
| 2-cluster model | 4204 | 16.5 | 190 414.9 | 190 352.8 | 0.90 |
| 3-cluster model | 3795 | 14.4 | 181 308.3 | 181 213.1 | 0.89 |
| **4-cluster model*** | **1099** | **4.3** | **179 517.1** | **179 388.9** | **0.87** |
| 5-cluster model | 1156 | 4.4 | 175 817.0 | 175 655.6 | 0.86 |
| 6-cluster model | 1134 | 4.4 | 175 451.2 | 175 256.7 | 0.86 |
| 7-cluster model | 688 | 3.7 | 17 117.5 | 173 890.0 | 0.79 |

*The models presented are shown in bold.
AIC, Akaike information criterion; APP, average posterior probability; BIC, Bayesian information criterion.

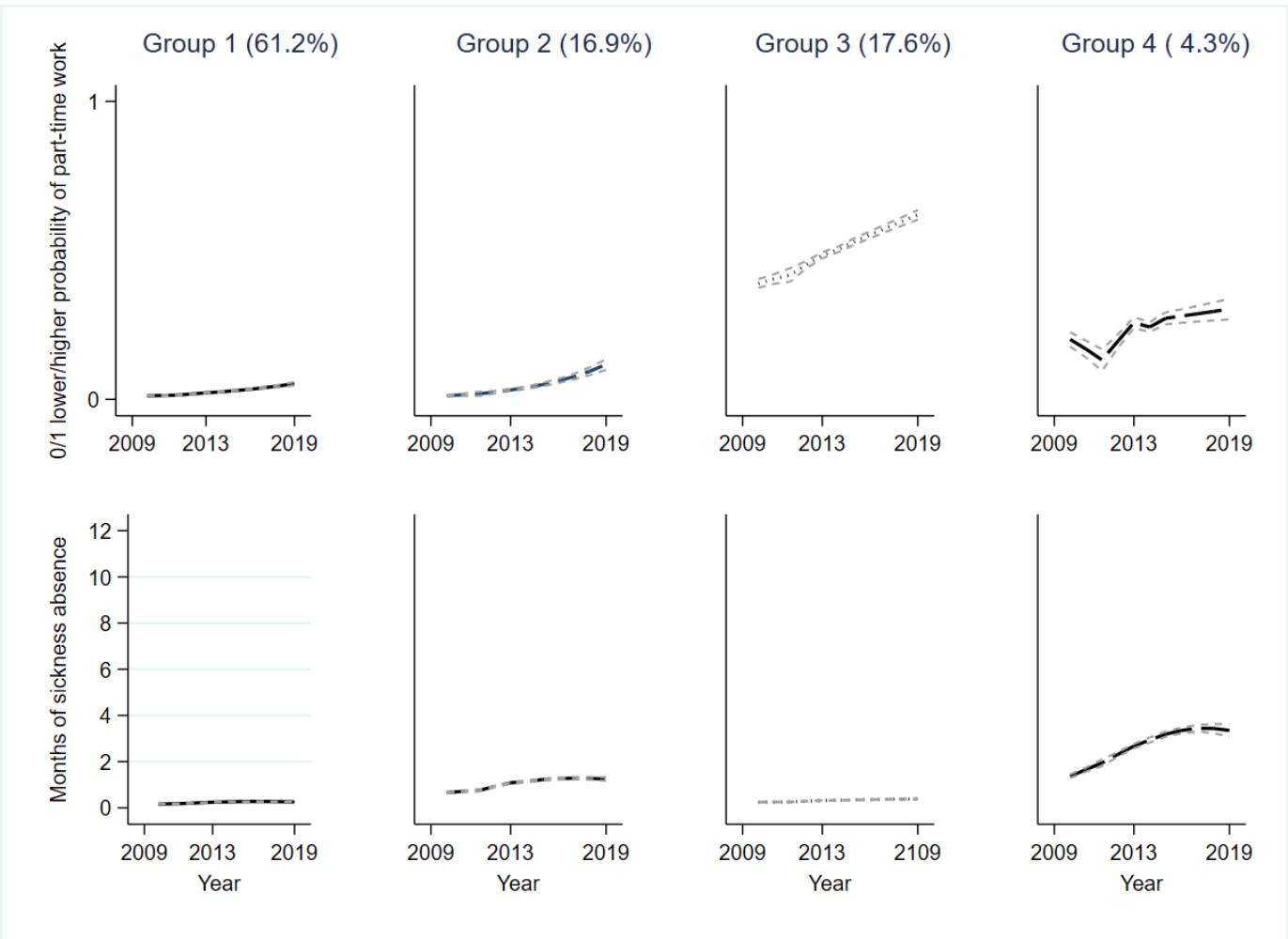

**Figure 2** Four clusters of trajectories of part-time work and months of sickness absence adjusted for time-variant night work (95% CIs are shown as dotted lines, but poorly visible due to being very narrow).

night work (online supplemental figure 1), and for each age group separately (online supplemental figures 2–5) and their fit indexes in the online supplemental tables 1 and 2. Although trajectory groups 1–3 are similar to adjusting for time-variant night work, group 4 indicates some changes in the probability of part-time work. Except for those aged 25–39 years with four trajectory groups, all other age groups were best fitted to a five-group trajectory

**Table 2** Sex and age group associated with cluster membership (part-time work and months of sickness absence)

|  | Group 2 Slowly increasing the probability of part-time work and low but mildly increasing sickness absence | | Group 3 Increasing part-time work and no sickness absence | | Group 4 Fluctuating but increasing part-time work and highest and increasing levels of sickness absence | |
|---|---|---|---|---|---|---|
|  | **RR*** | **95% CI** | **RR** | **95% CI** | **RR** | **95% CI** |
| Women | 1 | | 1 | | 1 | |
| Men | **0.74** | **0.63, 0.86** | **0.75** | **0.65, 0.87** | **0.49** | **0.34, 0.71** |
| Age <25 years | 1 | | 1 | | 1 | |
| 25–39 years | **1.32** | **1.13, 1.54** | **1.56** | **1.36, 1.80** | **2.36** | **1.56, 3.56** |
| 40–54 years | **1.83** | **1.58, 2.12** | 1.10 | 0.95, 1.26 | **3.79** | **2.54, 5.66** |
| ≥55 years | **1.69** | **1.39, 2.07** | **1.53** | **1.27, 1.85** | **2.99** | **1.83, 4.88** |

Multinomial regression (ref=cluster 1 'full-time work and no sickness absence', n=2124). RR ratio with 95% CI.
*Statistically significant RR with 95% CI in boldface.
RR, relative risk.

solution. The concurrent changes in part-time work and sickness absence varied between age groups: among the oldest age group, ≥55 years, three out of four trajectory groups had an increase in part-time work associated with low or no sickness absence. The concurrent trajectories with the decrease in part-time work (transition to full-time) with no sickness absence were identified among middle age groups (25–54 years).

## DISCUSSION

This longitudinal study of nearly 29 000 Finnish employees in the healthcare sector investigated the concurrent changes in part-time work and sickness absence across 11 years, and the role of age and sex on the identified concurrent trajectories. Four trajectory groups were detected: group 1 (61%) with full-time work and no sickness absence, group 2 (17%) with slowly increasing probability of part-time work and low but mildly increasing sickness absence, group 3 (18%) with increasing part-time work and no sickness absence, and group 4 (4%) with fluctuating but increasing part-time work and highest and increasing levels of sickness absence. Women and older employees were more likely to belong to groups 2–4 characterised by increasing part-time work. This is among the first studies to investigate the concurrent changes in part-time work and sickness absence from 2009 to 2019 while controlling for night work among healthcare employees. Hence, this study adds to the earlier knowledge based on survey data.[15 18–23]

The largest trajectory group (61%) worked full-time and had no sickness absence reflecting the well-known trend of women working mostly full-time in the Nordic countries.[11] However, we also detected two moderate size groups (trajectory groups 2 and 3) with increasing probability of part-time work and low but mildly increasing or no sickness absence. These might be indicative of our hypotheses that (a) older employees may choose part-time work due to decreasing health (with higher sickness absence), or (b) part-time work with low or no sickness absence might appear among younger employees due to the need to balance work and home life. Since the fourth trajectory group, although small (4%), indicated fluctuating, but increasing part-time work and highest and increasing sickness absence, it seems to support the above-mentioned hypothesis, and the assumption that both part-time work and sickness absence increase along with age. Our online supplemental figure 1 confirmed that overall, the four-group trajectory solution was the best even without time-variant night work. However, when night work was not accounted for (online supplemental figure 1), the probability of part-time work was less likely indicating that transitioning to day work may be another strategy to affect the working hours along age.[25] This might be an important notion for workplaces and occupational healthcare since age-related trends in the associations of night work and health exist[26 27] and perhaps modifications to working hours could promote

sustainable working life[14 15] if part-time work would be compensated with social benefits.

We were specifically interested in the role of sex and age on these concurrent changes of part-time work and sickness absence. Women had a higher likelihood of belonging to trajectory groups 2–4 characterised by increases in part-time work and with or without sickness absence. This is in line with earlier findings indicating that, for example, a typical employee having long-term sickness absence is a woman in her 50s and working in the healthcare sector.[37 38] This might be because women have higher sickness absence rates than men,[39–41] but also because women outnumber men in the healthcare sector (in our study, only 8% were men). Another influential, well-known fact is that men and women work part-time due to different reasons such as childcare or health[11] and even societal factors such as the availability of daycare for children or salary levels can play a role.[42–44] Older age groups had a higher likelihood of belonging to trajectory groups 2–4 compared with group 1. There are some possible explanations. First, the concurrent increase of part-time work and full-time sickness absence along with age. Alternatively, age-related trends of reacting to night work,[28 29] that is, across age groups, the concurrent changes of part-time work and sickness absence were different as shown in our online supplemental figures. Thus, further studies would be needed to address the various reasons for part-time work (eg, health, availability of work), health issues (diagnoses for sickness absence), but also more detailed periods (eg, monthly, or weekly) and working hour-based measures of part-time work. In addition, non-observational research designs are also needed.

We had access to objective working hour data including sickness absence from 2008 to 2019 almost in total 240 000 Finnish employees in the healthcare sector. This unique dataset enabled us to restrict the sample to those working shift work, having complete data for ≤3 consecutive years during 2009–2019 and without baseline ≥14 days of sickness absence. These are evident strengths, although the final sample size was around 29 000 employees. Such data are free from reporting or memory bias adding to earlier knowledge based on mainly survey data.[15 18–23] However, a limitation is that we had no information on the reasons for part-time work which may be due to, for example, health, personal matters or unavailability of full-time work.[45 46] We also cannot rule out the healthy worker bias but assume it might not play a major role. Furthermore, we only had information on sickness absence without diagnosis and no information on part-time sickness absence. Since part-time work may relate to health status, that is, being based on part-time sickness absence or disability pension,[9 10] detailed health information should be accounted for in further studies. We also lacked information on many sociodemographic factors (ie, marital status, number of children, level of income), which is a limitation. Further studies should be designed to address these factors. As in any observational study, we

were unable to demonstrate the causality between part-time work and sickness absence. It may be that transfer into part-time work indicates health problems and thus is a marker or a risk factor for subsequent sickness absence. However, part-time work may also be a stepping stone when returning to work after sickness absence. Future research on the inter-relations between these two phenomena is needed. Yet, another limitation might be the generalisability of the findings. In Finland, and Nordic countries, work in healthcare is organised mostly via irregular working hours, that is, shift work with a non-standard working hour schedule with variation in start and finish times, lengths and rest periods between work shifts.[47] On the other hand, sickness absence is subsidised via the welfare society model typical to Nordic countries. Hence, our findings might be more generalisable to Nordic than other countries.

## CONCLUSIONS

The majority (61%) of employees worked full-time and without sickness absence throughout the 11-year follow-up. However, for the rest of the employees, the probability of part-time work increased over time with varying levels of simultaneous changes in sickness absence. An increase in part-time work was more probable for women and older employees. For a minority of the employees (4%), the increase in part-time work was strongly linked with the concurrent increase in sickness absence. The relationship between part-time work and sickness absence is likely bidirectional. Transfer into part-time work may indicate health problems and subsequent sickness absence, but part-time work may also be used when gradually returning to work after a sickness absence and thus it can promote sustainable working life in healthcare employees.

**Contributors** AR, MH and JE contributed to the study's conception and design. Material preparation, data collection and analysis were performed by AR. The first draft of the manuscript was written by AR, and all the authors commented on previous versions of the manuscript. All the authors read and approved the final manuscript. AR and MH are responsible for the overall content as the guarantor.

**Funding** This work was supported by the Academy of Finland (DIGIHUM-programme grant number 329200).

**Competing interests** None declared.

**Patient and public involvement** Patients and/or the public were not involved in the design, or conduct, or reporting, or dissemination plans of this research.

**Patient consent for publication** Not required.

**Ethics approval** This study was fully based on employer-owned administrative employment register data that the hospital districts and cities had permitted access to and applied pseudonymised identification numbers for research purposes. Using such data does not include any experimental protocols requiring approval. In Finland, research using such data does not need to undergo review by an ethics committee according to national legislation (Medical Research Act).

**Provenance and peer review** Not commissioned; externally peer reviewed.

**Data availability statement** Data may be obtained from a third party and are not publicly available. The data that support the findings of this study are available from the hospital districts and cities but restrictions apply to the availability of these data, which were used under license for the current study, and so are not publicly

available. Data are however available from the last author upon reasonable request and with permission of the hospital districts and cities.

**ORCID iDs**
Annina Ropponen http://orcid.org/0000-0003-3031-5823
Jenni Ervasti http://orcid.org/0000-0001-9113-2428

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
