## [Reviewer comments · BMJ Open]

ARTICLE DETAILS

TITLE (PROVISIONAL)	Concurrent trajectories of part-time work and sickness absence - a longitudinal cohort study over 11 years among shift working hospital employees
AUTHORS	Ropponen, Annina; Ervasti, Jenni; Harma, Mikko

VERSION 1 – REVIEW

REVIEWER	Meagan Crowther Appleton Institute
REVIEW RETURNED	28-Jun-2023

GENERAL COMMENTS	The manuscript provides a sound description of the analysis of sickness absence and part time employment via the use of payroll data. BACKGROUND The background provides justification for why this study was conducted. The background would benefit from further details of why sickness absence is an important area to consider (i.e., does sickness absence cost the economy?) The sentence on Page 4, Line 24-29 "Part-time work, or a change from night work to day-time work, may provide a possibility to decrease workload due to demanding working conditions or to arrange an more time for care for small children or other dependents" is difficult to follow. Also the following sentence "Reasons of switching to part-time work may also be health-related, i.e., being based on part-time sickness absence (SA) or disability pension" Please consider whether sickness absence needs to be an abbreviation as it can be jarring for the reader, given it is not a common acronym. Page 5, Line 10-15 - What is author trying to say with this sentence "The possibility for part-time work, that is, to shorten working hours when having health problems and while compensated by social benefits, may extend and/or maintain the working lives of the older employees."? It is difficult to understand what point you are making. METHODS Please provide justification for "For this study, we restricted the sample to those with ≥ 31 work shifts/year in three consecutive years during 2008-2019." as those with extended sickness absence may work less than 31 shifts Well justified statistical analysis.
--

	Page 7, Line 32-33 ". In average, 7% worked part-time and even there were no ≥ 14 days SA" this sentence does not make sense - what is the author trying to convey? DISCUSSION The discussion provides interpretation of the results and highlights limitations and strengths of the present manuscript. The discussion would benefit from exploration of the findings between different age groups. For example, RR of trajectory 4 is higher in 40-54yrs than it is in >55 years. Why do you think this is? The discussion may be further enhanced with some detail of suggest future research. What research should come from your findings? OVERALL The manuscript would benefit from very careful proofread and review. There are many typos, language, tense and grammar errors throughout the manuscript which detract from the work.
--	--

REVIEWER	Helena Breth Nielsen National Research Centre for the Working Environment
REVIEW RETURNED	29-Jun-2023

GENERAL COMMENTS	Thank you for the opportunity to review this paper. The manuscript entitled "Concurrent trajectories of part-time work and full-time sickness absence among a cohort of shift working hospital employees" examines the patterns of part-time work and sickness absence among a cohort of hospital employees including 28 969 individuals. The authors identify four trajectory groups of part-time work and sickness absence and describe the sex and age characteristics of the groups. The aim is clear and interesting. The authors have used a novel approach in the field of part-time work and sickness absence and the inclusion of longitudinal register-based data is a major strength of this study. Moreover, the article is well-structured and written. I find the manuscript suitable for publication with some smaller revisions. Please find my suggestions below. Title: - The term "full-time sickness absence" can be a bit confusing in the title as "full-time" could also relate to part-time work. Consider renaming the title e.g.: "Concurrent trajectories of part-time work and sickness absence among a cohort of shift working hospital employees". Abstract: - On page (p) 2, line (l) 21: Please clarify the sentence: "Lack of baseline SA (≥ 14 days), three consecutive years of data and working shifts were set for final sample." - On p. 2, l. 55: "Other solutions than part-time work might merit attention to promote sustainable working life among health care employees". This insinuates that part-time work is not related to sickness absence, which I find is a strong conclusion based on the analyses. Please reconsider this statement or elaborate more in the discussion to support this. Strengths and limitations of the study:
--

- Will a part-time worker who gets too sick to work be registered with “full-time sickness absence” or “part-time sickness absence” due to their prior part-time employment relation? If they are registered as “part-time sickness absence”, then please mention the lack of information on part-time sickness absence here.

Introduction:

- How common are part-time work and sickness absences in Finland?
- The section on hypothesis on p. 5. Could benefit from more arguments on why these hypotheses arise e.g. linking the reasons for part-time work with the hypotheses.
- It may improve the manuscript if methodological choices are not stated in the aim, and instead listed in the method section., i.e. remove: “without SA (≥ 14 subsequent days) at the baseline while controlling the amount of night work”.

Methods:

- How is part-time work defined in terms of the number of hours? And what number of hours would be considered full-time work in Finland?
- On page. 5, line 57: It states that the population include those, who use Titania® shift scheduling software. Do you have any information on if there is some selection in who uses this software?
- Please state why you adjust for shift work (with night work as a proxy).
- Why is the short sickness absence (≤ 3 days) estimated in 2008 (baseline), when follow-up first started in 2009?
- The data is aggregated into yearly information. Why not use more detailed data (e.g., Daily or monthly)?
- Please consider, if the method used is actually “group-based multi-trajectory modelling” (Nagin et al. 2018).
- In the flow chart. Please add the number and % of excluded individuals.
- Do you have information on if part-time work is voluntary or involuntary? If so, I think it would be interesting to see analyses that included this aspect.

Results:

- Consider placing “Table 1 Goodness of fit statistics of group-based trajectory analysis models” in the appendix.
- Consider presenting the main graphs with the trajectories in the results section.
- Could the four trajectory groups be given a more informative name? E.g. constant full-time work and no sickness absence (“full-time/no SA”). Group 1-4 is not very informative and this makes it more difficult to follow the interpretation. Otherwise, I appreciate the effort to describe the main characteristics of the groups when interpreting the results. There are a few places where it is lacking (e.g. p. 8, l. 20; p. 9, l. 39; p.10, l. 44), please be consistent.
- On p. 8, l.2-15: After each group number there is a parenthesis with a percentage, which I assume is the percentage of the total population. Please add what the percentage refers to.
- Are the numbers in Table 2 used to profile the trajectory groups in terms of age and sex? Please consider clarifying this.

Discussion:

- Consider if it is possible to combine the discussion of the main trajectory findings and the findings on age and sex. Instead of first discussing the main analyses reflecting on gender and age, and

	then afterwards discussing the analyses of sex and age characteristics of the groups, this could be collapsed into the same paragraph.  - On p. 10, l. 10: “Our Supplemental figure S1 indicated that overall, four group trajectory solution was the best even without time variant night work, but the probability of part-time work was less likely indicating that transition to day work may be another strategy to affect the working hours along age”. I am not following the argument here, please rephrase this sentence. - I valued the reflections of characteristics on gender and age and p. 10, l. 38. However, I missed some more reflection on explanations of the identified groups based on their trajectory shapes. They likely reflect different reasons for part-time work e.g. having small children, having multiple jobs, no vacant full-time position, health-related, taking care of close relatives, divorce or other unexpected events. - It would be valuable to include a discussion about the more descriptive nature of the analyses with regard to the interpretation of the findings e.g. in terms of possible confounding from having children, disability or SES. - In addition, part-time work and sickness absence are measured concurrently and aggregated on a yearly basis. Therefore, I think it is relevant to discuss reverse causality as part-time work could be the step before sickness-absence, as well as a stepping stone out of sickness-absence. - Could the inclusion criteria of employment for three years (p. 5, l. 50: “≥31 work shifts/year in three consecutive years”) introduce bias, due to a healthy worker effect? Conclusion:  - I missed a more clear description of the main findings. E.g., “In this paper, four distinct patterns of part-time work and concurrent sickness absence were identified”. - On p. 11, l. 43: Please be specific instead of using “these” in the sentence. - In the conclusion it states “Other working hour solutions than part-time work might merit attention to promote sustainable working life among health care employees.” I would like some more arguments in the discussion to support this, as I think it insinuates no association between part-time work and sickness absence, which is not directly analysed. Generally  - There are a few missing or misplaced words throughout the paper, please check for this (e.g. p. 4, l. 17, delete “for”). - Please use space or no space consistently throughout the paper, when writing large numbers e.g. 28969 vs. 28 969 health care employees (e.g. in the flow chart vs. p.9, l. 21). - Please write out “and” or “or” instead of “/” (e.g. p. 7, l. 41). - In the trajectories, please label the X and Y-axis consistently with the trajectories in the supplementary material. References:  - Nagin DS, Jones BL, Passos VL, Tremblay RE. Group-based multi-trajectory modeling. Statistical Methods in Medical Research. 2018;27(7):2015-2023. doi:10.1177/0962280216673085
--	---

VERSION 1 – AUTHOR RESPONSE

Reviewer: 1

Ms. Meagan Crowther, Appleton Institute

Comments to the Author:

The manuscript provides a sound description of the analysis of sickness absence and part time employment via the use of payroll data.

Response: We appreciate this overall positive review on our manuscript.

BACKGROUND

The background provides justification for why this study was conducted. The background would benefit from further details of why sickness absence is an important area to consider (i.e., does sickness absence cost the economy?)

Response: On page 4, the end of the paragraph 1, we amended: **As sickness absence increases costs to both the employers and the society, further understanding of the linkage with part-time work is needed.**

The sentence on Page 4, Line 24-29 "Part-time work, or a change from night work to day-time work, may provide a possibility to decrease workload due to demanding working conditions or to arrange an more time for care for small children or other dependents" is difficult to follow. Also the following sentence "Reasons of switching to part-time work may also be health-related, i.e., being based on part-time sickness absence (SA) or disability pension".

Response: These sentences have been revised to improve clarity, as follows: **In shift work, two working hour solutions may provide a possibility to decrease workload due to demanding working conditions, or to arrange more time for care for small children or other dependents: an employee may work part-time, or to change from night work to day-time work^{7 8}. However, part-time work may also be health-related, i.e., being based on part-time sickness absence or disability pension^{9 10}.**

Please consider whether sickness absence needs to be an abbreviation as it can be jarring for the reader, given it is not a common acronym.

Response: We have removed the use of SA (abbreviation for sickness absence) from the main text to add clarity. However, the abbreviation was kept in the abstract due to word limitation.

Page 5, Line 10-15 - What is author trying to say with this sentence "The possibility for part-time work, that is, to shorten working hours when having health problems and while compensated by social benefits, may extend and/or maintain the working lives of the older employees."? It is difficult to understand what point you are making.

Response: We apologize for the unclarity. The sentence reads now: The possibility **to combine part-time work with part-time sickness absence**, that is, to shorten working hours when having health problems compensated by social benefits (**i.e., part-time sickness absence**), may extend and/or maintain the working lives of the older employees²⁴.

METHODS

Please provide justification for "For this study, we restricted the sample to those with ≥31 work shifts/year in three consecutive years during 2008-2019." as those with extended sickness absence may work less than 31 shifts.

Response: Very relevant request and we have amended the sentence with **"To ensure follow-up across years and to exclude those with exceptionally long absences (due to sickness, parental leave, or other reasons) and very short working periods, we restricted the sample to those with ≥31 work shifts/year in three consecutive years during 2008-2019."**

Well justified statistical analysis.

Response: Thank you!

Page 7, Line 32-33 ". On average, 7% worked part-time and even there were no ≥ 14 days SA" this sentence does not make sense - what is the author trying to convey?

Response: We have revised the sentence as follows: "**At baseline in 2008, 7% of the final sample of 28969 shift working employees worked part-time, mean age was 39.8 years (SD 10.7), 92% were women, and 34% had had at least one short-term sickness absence (≤ 3 days).**"

DISCUSSION

The discussion provides interpretation of the results and highlights limitations and strengths of the present manuscript.

Response: Thank you for the positive feedback.

The discussion would benefit from exploration of the findings between different age groups. For example, RR of trajectory 4 is higher in 40-54yrs than it is in >55 years. Why do you think this is?

Response: We appreciate this suggestion. Although we agree with the reviewer that there are differences in point estimates (RR) for age groups, we would like to retain the modest interpretations of them as the point estimates are always dependent on the sample (e.g., size of the groups). Furthermore, we present the age-group specific trajectory groups in supplementary material for interested readers. As stated in our original version of the manuscript on page 10 (end of the paragraph 2), there were different concurrent changes in part-time work and sickness absence potentially reflecting health-, life-, or work-related differences. To address this, we added a sentence to the discussion on page 10, at the end of the paragraph 2: "**Thus, further studies would be needed to address the various reasons of part-time work (e.g., health, availability of work), health issues (diagnoses for sickness absence), but also more detailed time periods (e.g., monthly, or weekly) and working hour-based measures of part-time work.**"

The discussion may be further enhanced with some detail of suggest future research. What research should come from your findings?

Response: We appreciate this suggestion and as indicated in our previous response above, we considered that: "**Thus, further studies would be needed to address the various reasons of part-time work (e.g., health, availability of work), health issues (diagnoses for sickness absence), but also more detailed time periods (e.g., monthly, or weekly) and working hour-based measures of part-time work. In addition, non-observational research designs are also needed.**"

OVERALL

The manuscript would benefit from very careful proofread and review. There are many typos, language, tense and grammar errors throughout the manuscript which detract from the work.

Response: A careful proof reading has been performed for this manuscript.

Reviewer: 2

Dr. Helena Breth Nielsen, National Research Centre for the Working Environment

Comments to the Author:

Thank you for the opportunity to review this paper. The manuscript entitled “Concurrent trajectories of part-time work and full-time sickness absence among a cohort of shift working hospital employees” examines the patterns of part-time work and sickness absence among a cohort of hospital employees including 28 969 individuals. The authors identify four trajectory groups of part-time work and sickness absence and describe the sex and age characteristics of the groups.

The aim is clear and interesting. The authors have used a novel approach in the field of part-time work and sickness absence and the inclusion of longitudinal register-based data is a major strength of this study. Moreover, the article is well-structured and written. I find the manuscript suitable for publication with some smaller revisions. Please find my suggestions below.

Response: We thank the reviewer for this positive review on our manuscript.

Title:

- The term “full-time sickness absence” can be a bit confusing in the title as “full-time” could also relate to part-time work. Consider renaming the title e.g.: “Concurrent trajectories of part-time work and sickness absence among a cohort of shift working hospital employees”.

Response: We have reviewed the title based on this and accounting for the comments from the journal. The revised title reads: **Concurrent trajectories of part-time work and sickness absence - a longitudinal cohort study over 11 years among shift working hospital employees**

Abstract:

- On page (p) 2, line (l) 21: Please clarify the sentence: “Lack of baseline SA (≥ 14 days), three consecutive years of data and working shifts were set for final sample.”

Response: The sentence reads now:” **The final sample included those working shifts with three consecutive years of data and without baseline (≥ 14 days) SA.”**

- On p. 2, l. 55: “Other solutions than part-time work might merit attention to promote sustainable working life among health care employees”. This insinuates that part-time work is not related to sickness absence, which I find is a strong conclusion based on the analyses. Please reconsider this statement or elaborate more in the discussion to support this.

Response: We agree with the reviewer and the sentence has been revised to: **“Part-time work, but also other solutions** might merit attention to promote sustainable working life among health care employees.” We revised the statement in the manuscript conclusions (page 12) as well.

Strengths and limitations of the study:

- Will a part-time worker who gets too sick to work be registered with “full-time sickness absence” or “part-time sickness absence” due to their prior part-time employment relation? If they are registered as “part-time sickness absence”, then please mention the lack of information on part-time sickness absence here.

Response: We had no information about part-time vs. full-time sickness absence, hence we added the lack of information on part-time sickness absence here.

Introduction:

- How common are part-time work and sickness absences in Finland?

Response: 10% of all employed have sickness absences and around 20% of women (13% of men) work part-time in Finland. We have added this information to page 4 with relevant references.

- The section on hypothesis on p. 5. Could benefit from more arguments on why these hypotheses arise e.g. linking the reasons for part-time work with the hypotheses.

Response: We have followed this suggestion and revised the text on the hypothesis section.

- It may improve the manuscript if methodological choices are not stated in the aim, and instead listed in the method section., i.e. remove: "without SA (≥ 14 subsequent days) at the baseline while controlling the amount of night work".

Response: Corrected as suggested.

Methods:

- How is part-time work defined in terms of the number of hours? And what number of hours would be considered full-time work in Finland?

Response: We appreciate these questions. However, in this study, we utilized the work contract information on part-time work (i.e., < 100% of working time) that was available from the daily working hour data (as indicated on page 6, paragraph 2). Hence, we had the daily information if the employee had had part-time work contract (<100% of work time) or not (= 100% working time) and that was used to estimate the part-time work (without any specific working hour limits). This decision was made because working hours in the health care sector are irregular due to period-based planning system reflecting the needs of 24/7 operating hours. The period-based working hours are planned for 3-week periods which means there are fluctuations in (weekly) working hours despite the work contract. However, in full-time work, the 3-week period working hours should be set to 114 hours 45 minutes within the adjustment period (than can vary between 6-12 weeks), so that would have been the definition of full-time work in our sample if that would have been used. In future studies, an approach based on the realized working hours could be utilized to shed further light to the details of working hours in part-time vs. full-time work. To appreciate the reviewer's opinion, this need for further studies was added to the discussion section on page 10, at the end of the paragraph 2.

- On page. 5, line 57: It states that the population include those, who use Titania® shift scheduling software. Do you have any information on if there is some selection in who uses this software?

Response: In Finland, before the healthcare reform in 2023, there were 21 hospital districts. Our data comprises 10 of them including the four largest hospital districts. Together with data from cities, our data should be well representative of Finnish public healthcare sector. Titania® shift scheduling software is the most common software for shift scheduling in the Finnish public healthcare sector, and it is used approximately by 95% of the entire public healthcare sector. Inside the hospital districts, all employees having the period-based work contract must use the software due to payroll.

- Please state why you adjust for shift work (with night work as a proxy).

Response: We added on page 6, paragraph 2: "**Since aging employees may change from night work to day work^{25 34 35}, the proportion of night work was assumed to reflect this change, which in turn might affect the rate of part-time work.**"

- Why is the short sickness absence (≤ 3 days) estimated in 2008 (baseline), when follow-up first started in 2009?

Response: Previous sickness absence in the strongest predictor for sickness absence (see e.g., Gohar B, Larivière M, Lightfoot N, Larivière C, Wenghofer E, Nowrouzi-Kia B. Demographic, Lifestyle, and Physical Health Predictors of Sickness Absenteeism in Nursing: A Meta-Analysis. *Saf Health Work.* 2021 Dec;12(4):536-543. doi: 10.1016/j.shaw.2021.07.006., or Laaksonen M, He L, Pitkäniemi J. The durations of past sickness absences predict future absence episodes. *J Occup Environ Med.* 2013 Jan;55(1):87-92. doi: 10.1097/JOM.0b013e318270d724.). Although we limited the sample to those who did not have any long (≥ 14 consequent days) sickness absence in baseline, we wanted to control for short-term sickness absence.

- *The data is aggregated into yearly information. Why not use more detailed data (e.g., Daily or monthly)?*

Response: We fully agree with the reviewer that more detailed data might be very relevant. However, since we utilized 11 years of follow-up and selected the group-based multi-trajectory modelling (this confirms that the reviewer is also right in her next comment) for analyses, using more detailed data would have added time points and complicated the analysis. Furthermore, as this is among the first studies to investigate concurrent changes in part-time work and sickness absence, we considered the annual data relevant for exploring the phenomena. However, to acknowledge this consideration, we would like to mention that we have considered more detailed data as one for the future needs of studies in our discussion on page 10, at the end of the paragraph 2.

- *Please consider, if the method used is actually “group-based multi-trajectory modelling” (Nagin et al. 2018).*

Response: Yes, the reviewer is right. Thank you for pointing us to this typo.

- *In the flow chart. Please add the number and % of excluded individuals.*

Response: Done as suggested.

- *Do you have information on if part-time work is voluntary or involuntary? If so, I think it would be interesting to see analyses that included this aspect.*

Response: Unfortunately, we lack that information.

Results:

- *Consider placing “Table 1 Goodness of fit statistics of group-based trajectory analysis models” in the appendix.*

Response: Thank you for the suggestion, but we decided to keep the table as it is essential for understanding the basis of the trajectory model selection. Since we do not have that many tables, we assumed it might fit too.

- *Consider presenting the main graphs with the trajectories in the results section.*

Response: Yes, the Figure 2 (please note that the numbering of the figures has been changed and the flow chart in the methods section is Figure 1) includes the main graphs.

- *Could the four trajectory groups be given a more informative name? E.g. constant full-time work and no sickness absence (“full-time/no SA”). Group 1-4 is not very informative and this makes it more difficult to follow the interpretation. Otherwise, I appreciate the effort to describe the main characteristics of the groups when interpreting the results. There are a few places where it is lacking (e.g. p. 8, l. 20; p. 9, l. 39; p. 10, l. 44), please be consistent.*

Response: Thank you for this suggestion. We assumed that the reviewer wished more informative names for the trajectory groups in the figure (currently Figure 2)? We tried to put the names but that would have resulted use of very, very small font and reducing the graph sizes. However, the trajectory names are listed on page 7, and we would expect the journal to place the Figure 2 near them. Also, the trajectory names are as full in the Table 2. Furthermore, we have cross-checked the trajectory names for consistency in the text.

- On p. 8, l.2-15: After each group number there is a parenthesis with a percentage, which I assume is the percentage of the total population. Please add what the percentage refers to.

Response: Added as suggested when the first % is mentioned.

- Are the numbers in Table 2 used to profile the trajectory groups in terms of age and sex? Please consider clarifying this.

Response: The numbers in Table 2 are risk ratios (RR) with 95% confidence intervals (CI) for trajectory groups 2-4 in comparison to group 1 and in sex and age categories as indicated in the rows. Hence, they indicate the relative probability of men, women, and different age groups for belonging to each trajectory group. This has been indicated in the text referring to Table 2.

Discussion:

- Consider if it is possible to combine the discussion of the main trajectory findings and the findings on age and sex. Instead of first discussing the main analyses reflecting on gender and age, and then afterwards discussing the analyses of sex and age characteristics of the groups, this could be collapsed into the same paragraph.

Response: Thank you for this suggestion. We modified the synopsis of results as follows: **This longitudinal study of nearly 29 000 Finnish employees in health care sector investigated the concurrent changes in part-time work and sickness absence across 11 years, and the role of age and sex on the identified concurrent trajectories. Four trajectory groups were detected: Group 1 (61%) with full-time work and no sickness absence, Group 2 (17%) with slowly increasing probability of part-time work and low but mildly increasing sickness absence, Group 3 (18%) with increasing part-time work and no sickness absence, and Group 4 (4%) with fluctuating but increasing part-time work and highest and increasing levels of sickness absence. Women and older employees were more likely to belong to groups 2-4 characterized by increasing part-time work.**

- On p. 10, l. 10: "Our Supplemental figure S1 indicated that overall, four group trajectory solution was the best even without time variant night work, but the probability of part-time work was less likely indicating that transition to day work may be another strategy to affect the working hours along age". I am not following the argument here, please rephrase this sentence.

Response: We agree with the reviewer, the sentence missed a specification that has been added now on page 10, paragraph 1, as follows: "Our Supplemental figure S1 **confirmed** that overall, four group trajectory solution was the best even without time variant night work. **However, when night work was not accounted for (Supplemental figure S1), the probability...**"

- I valued the reflections of characteristics on gender and age and p. 10, l. 38. However, I missed some more reflection on explanations of the identified groups based on their trajectory shapes. They likely reflect different reasons for part-time work e.g. having small children, having multiple jobs, no vacant full-time position, health-related, taking care of close relatives, divorce or other unexpected events.

Response: We appreciate the point made by the reviewer. However, since we lacked information on these issues, we tried to be modest in our interpretations and explanations. As we have identified the need for more detailed information of both part-time work and sickness absence as a focus for future studies (in the discussion on page 10, at the end of the page), we would like to keep the discussion as it is.

- It would be valuable to include a discussion about the more descriptive nature of the analyses with regard to the interpretation of the findings e.g. in terms of possible confounding from having children, disability or SES.

Response: To address this, we added few sentences to the discussion on page 11, paragraph 1: **“We also lacked information on many sociodemographic factors (i.e., marital status, number of children, level of income) which is a limitation. Further studies should be designed to address these factors.”**

- In addition, part-time work and sickness absence are measured concurrently and aggregated on a yearly basis. Therefore, I think it is relevant to discuss reverse causality as part-time work could be the step before sickness-absence, as well as a stepping stone out of sickness-absence.

Response: Yes, the reviewer is right. However, group-based multi-trajectory modelling focuses on concurrent changes within the trajectories of two phenomena. In this manuscript, we have tried to avoid any discussion on cause-effect-association as that has not directly been analyzed. We have added this to limitations and further research needs on page 11, paragraph 1, as follows: **“As in any observational study, we were unable to demonstrate the causality between part-time work and sickness absence. It may be that transfer into part-time work indicates health problems and thus is a marker or a risk factor for subsequent sickness absence. However, part-time work may also be a steppingstone when returning to work after sickness absence. Future research on the interrelations between these two phenomena is needed.”**

- Could the inclusion criteria of employment for three years (p. 5, l. 50: “≥31 work shifts/year in three consecutive years”) introduce bias, due to a healthy worker effect?

Response: We agree with the reviewer that “healthy worker effect” may play a role. However, the turnover rate is rather high in healthcare sector (see e.g., Krutova O, Peutere L, Ervasti J, Härmä M, Virtanen M, Ropponen A. Sequence analysis of the combinations of work shifts and absences in health care - comparison of two years of administrative data. BMC Nurs. 2022 Dec 30;21(1):376. doi: 10.1186/s12912-022-01160-1.) and as indicated before, we lacked the knowledge of reasons for part-time work. Furthermore, we restricted the sample to those without long (≥14 consequent days) sickness absence in baseline (i.e., being healthy at baseline), the limit of one month (i.e., ≥31 work shifts/year in total) at work during 3 consequent years is rather modest restriction. Therefore, we assume that the healthy worker bias cannot be ruled out but not to play a major role. However, this criterion was necessary to assure relevant follow-up across 11 years due to above mentioned high turnover. We have addressed the healthy worker effect in the discussion, on page 11, paragraph 1.

Conclusion:

- I missed a more clear description of the main findings. E.g., “In this paper, four distinct patterns of part-time work and concurrent sickness absence were identified”.

Response: Thank you, following this suggestion we made a full revision for the conclusions. They read now starting from the end of the page 11: **The majority (61%) of employees worked full-time and without sickness absence throughout the 11-year follow-up. However, for the rest of the employees, the probability of part-time work increased over time with varying level of simultaneous changes in sickness absence. Increase in part-time work was more probable for women and for older employees. For a minority of the employees (4%), the increase in part-**

time work was strongly linked with concurrent increase in sickness absence. The relationship between part-time work and sickness absence is likely bidirectional. Transfer into part-time work may indicate health problems and subsequent sickness absence, but part-time work may also be used when gradually returning to work after sickness absence and thus it can promote sustainable working life in healthcare employees.

- On p. 11, l. 43: Please be specific instead of using “these” in the sentence.

Response: We have corrected the sentence as suggested.

- In the conclusion it states “Other working hour solutions than part-time work might merit attention to promote sustainable working life among health care employees.” I would like some more arguments in the discussion to support this, as I think it insinuates no association between part-time work and sickness absence, which is not directly analysed.

Response: Thank you, we have edited the conclusions accordingly, and the discussion has been amended based on the suggestions of 2 reviewers that will also grasp this. See e.g., page 11.

Generally

- There are a few missing or misplaced words throughout the paper, please check for this (e.g. p. 4, l. 17, delete “for”).

Response: We have carefully edited the text, thanks for noting.

- Please use space or no space consistently throughout the paper, when writing large numbers e.g. 28969 vs. 28 969 health care employees (e.g. in the flow chart vs. p.9, l. 21).

Response: Corrected as suggested (throughout the manuscript).

- Please write out “and” or “or” instead of “/” (e.g. p. 7, l. 41).

Response: Done.

- In the trajectories, please label the X and Y-axis consistently with the trajectories in the supplementary material.

Response: Done.

References:

- Nagin DS, Jones BL, Passos VL, Tremblay RE. Group-based multi-trajectory modeling. *Statistical Methods in Medical Research*. 2018;27(7):2015-2023. doi:10.1177/0962280216673085

Response: Thanks!

VERSION 2 – REVIEW

REVIEWER	Meagan Crowther Appleton Institute
REVIEW RETURNED	23-Aug-2023
GENERAL COMMENTS	Thank you for your careful revisions.